

# An automated, high-throughput method for standardizing image color profiles to improve image-based plant phenotyping

Jeffrey C. Berry, Noah Fahlgren, Alexandria A. Pokorny, Rebecca S. Bart and Kira M. Veley

Donald Danforth Plant Science Center, Saint Louis, MO, United States of America

## ABSTRACT

High-throughput phenotyping has emerged as a powerful method for studying plant biology. Large image-based datasets are generated and analyzed with automated image analysis pipelines. A major challenge associated with these analyses is variation in image quality that can inadvertently bias results. Images are made up of tuples of data called pixels, which consist of R, G, and B values, arranged in a grid. Many factors, for example image brightness, can influence the quality of the image that is captured. These factors alter the values of the pixels within images and consequently can bias the data and downstream analyses. Here, we provide an automated method to adjust an image-based dataset so that brightness, contrast, and color profile is standardized. The correction method is a collection of linear models that adjusts pixel tuples based on a reference panel of colors. We apply this technique to a set of images taken in a high-throughput imaging facility and successfully detect variance within the image dataset. In this case, variation resulted from temperature-dependent light intensity throughout the experiment. Using this correction method, we were able to standardize images throughout the dataset, and we show that this correction enhanced our ability to accurately quantify morphological measurements within each image. We implement this technique in a high-throughput pipeline available with this paper, and it is also implemented in PlantCV.

## INTRODUCTION

Imaging and computer vision approaches are powerful tools for plant phenotyping because they allow plant physical and physiological features to be measured non-destructively with relatively high quantitative and temporal resolution (*Fahlgren, Gehan & Baxter, 2015*). In addition to robotic platforms, micro-computers are making large scale image acquisition more affordable and accessible (*Tovar et al., 2018*). From these datasets, it is possible to measure plant size, shape, color, and other features in an automated fashion and correlate phenotypes with experimental treatments (*Honsdorf et al., 2014*; *Chen et al., 2014*; *Neilson et al., 2015*; *Fahlgren et al., 2015*; *Acosta-Gamboa et al., 2016*; *Feldman et al., 2017*; *Veley et al., 2017*; *Liang et al., 2018*).

Corresponding author
Kira M. Veley,
kveley@danforthcenter.org

The use of high-throughput imaging and computer vision analysis for plant phenotyping presents challenges in data acquisition, management, and analysis that require a team with diverse expertise in biology, engineering, and mathematics (*Furbank & Tester, 2011*; *Fahlgren, Gehan & Baxter, 2015*). One of these challenges is assuring standardization of methods so proper inferences can be made (*Wang & Zhang, 2010*; *Furbank & Tester, 2011*). For example, variation within an image dataset can have a significant impact on phenotype inferences and so must be considered. For a given pixel that is stored in an image there are three values, one for each component (red, green, and blue; RGB) (*Finlayson, Mackiewicz & Hurlbert, 2015*; *Gunturk et al., 2005*). Image brightness is an overview of how large the values are for each pixel, and image contrast is defined as the range of pixel values. Color profile is the range of colors observed in the image. What we perceive as image quality is a combination of contrast and color profile among other features (*Livingstone & Hubel, 1988*). Images with large contrast and color profile are considered high-quality images because they have larger numerical range relative to low-quality images.

Once an image set has been collected, image analysis is the process of extracting numerical data that describes the object in the image. First, the object must be separated from background pixels through a process called object segmentation. Accuracy of segmentation decreases as the image quality decreases. Various methods for segmentation have been proposed to handle varying image qualities including use of adaptive thresholding or learning algorithms (*Yogamangalam & Karthikeyan, 2013*). Rather than relying on mathematical models to segment the object in the image, we propose a method of image standardization based on a reference color palette and then altering the image so fixed-threshold segmentation becomes highly robust to image quality. Standardizing the image does not require a training set which is a major bottleneck in creating learning algorithms for segmentation and reduces the variability of autothresholding. We tested this method using a dataset from an automated imaging system containing approximately 24,000 images taken over an 11-day period. Each image contains a single plant and a reference color palette. Using a fixed-threshold segmentation routine to isolate the plant, various shapes and color profile were recorded for every image. We compare the quantification of plant features before and after image standardization and demonstrate that our method improves overall accuracy of image analysis.

## MATERIALS AND METHODS

### Software

Image analysis was performed with the C++ source file included with this manuscript and must be compiled against OpenCV (only tested on 3.1). Statistical analyses and graphics were done using R version 3.4.4 (*R Core Team, 2018*) with the following packages: ggplot 2.2.1, ggthemes 3.4.0, reshape2 1.4.3, plyr 1.8.4, grid 3.4.4, gridextra 2.2.1, vegan 2.4-6. $T$-test, and Wilcoxon signed-rank test were done with base R functions. Constrained analysis of principal coordinates was done with vegan.

## Plant growth and imaging conditions

The plant growth phenotyping data used in this experiment is a part of a large dataset. Plant growth conditions were based on a previous experiment (*Veley et al., 2017*) with adjustments. Details are as follows: Two sorghum genotypes (*Sorghum bicolor* (L.) Moench, genotypes BTx623 and China 17) were germinated between pieces of damp filter paper in petri dishes at 30 °C for 3 days prior to planting. The genotype information was collapsed throughout this analysis. Square pots (7.5 cm wide, 20 cm high) fitted with drainage trays were pre-filled with Profile Field & Fairway$^{TM}$ calcined clay mixture (Hummert International, Earth City, Missouri). On day 4 (4 DAP), germinated seeds were transplanted into pre-moistened pots, barcoded (including genotype identification, treatment group, and a unique pot identification number), randomized and scanned onto the Bellwether Phenotyping System (Conviron, day/night temperature: 32 °C/22 °C, day/night humidity: 40%/50%, day length: 14 hr (ZT 0–14 = hr 0,700–2,100), night length: 10 hr (ZT 14–24 = hr 2,100–0,700), light source: metal halide and high pressure sodium, light intensity: 400 $\mu$mol/m$^2$/s). Plants received 40 mL fertilizer treatments (see below) daily, and were watered to weight (640 g starting dry weight, 1,070 g target weight DAP 4–11, 960 g DAP 12–25, not including weight of phenotyping system carrier) daily by the system using distilled water. Fertilizer treatment information was collapsed for all analysis except where indicated, and were as follows:

### Fertilizer treatments

Control fertilizer treatment (100% N): 6.5 mM $KNO_3$, 4.0 mM $Ca(NO_3)_2 4H_2O$, 1.0 mM $NH_4H_2PO_4$, 2.0 mM $MgSO_4 7H_2O$, micronutrients, pH 4.6.

Low nitrogen fertilizer treatment (10% N): 0.65 mM $KNO_3$, 4.95 mM KCl, 0.4 mM $Ca(NO_3)_2 4H_2O$, 3.6 mM $CaCl_2 2H_2O$, 0.1 mM $NH_4H_2PO_4$, 0.9 mM $KH_2PO_4$, 2.0 mM $MgSO_4 7H_2O$, micronutrients, pH 5.0.

The same micronutrients were used for both treatments: 4.6 $\mu$M $H_3BO_3$, 0.5 $\mu$M $MnCl_2 4H_2O$, 0.2 $\mu$M $ZnSO_4 7H_2O$, 0.1 $\mu$M $(NH_4)_6 Mo_7O_{24} 4H_2O$, 0.2 $\mu$M $CuSO_4 5H_2O$, 71.4 $\mu$M Fe-EDTA.

### Standardization method

The image standardization method used here is based on a color transfer method that can adjust the colors in an image to match a target image color profile (*Gong, Finlayson & Fisher, 2016*). The goal is to create a transform such that when applied to the values of every pixel in a source image, it returns values mapped to a target image profile. It was shown that the color transfer is a single homography from source to target (*Gong, Finlayson & Fisher, 2016*). Hue-preserving methods have been characterized in detail and may be sufficient for some datasets (*Mackiewicz, Andersen & Finlayson, 2016*). However, we aimed to change the hue profile to eliminate hue bias introduced by light source quality batch effects. Color data from the source and the target image are used to define the homography and the color data must be sampled from homologous points, so a reference is included in the images from which the values of R, G, and B can be measured. We included a ColorChecker Passport

Photo (X-Rite, Inc.), which has a panel of 24 industry standard color reference chips, within each image. A target image (or reference) is declared and the process of computing the homography and applying it to a given source image is as follows:

1. Let $T$ and $S$ be matrices containing measurements for the R, G, and B components of each of the ColorChecker reference chips pictured in the target image and source image respectively.

$$T = \begin{bmatrix} T_R & T_G & T_B \end{bmatrix} = \begin{bmatrix} t_{r_1} & t_{g_1} & t_{b_1} \\ t_{r_2} & t_{g_2} & t_{b_2} \\ \vdots & \vdots & \vdots \\ t_{r_{24}} & t_{g_{24}} & t_{b_{24}} \end{bmatrix} \text{ and } S = \begin{bmatrix} S_R & S_G & S_B \end{bmatrix} = \begin{bmatrix} s_{r_1} & s_{g_1} & s_{b_1} \\ s_{r_2} & s_{g_2} & s_{b_2} \\ \vdots & \vdots & \vdots \\ s_{r_{24}} & s_{g_{24}} & s_{b_{24}} \end{bmatrix} \quad (1)$$

2. Extend $S$ to include the square and cube of each element.

$$S = \begin{bmatrix} s_{r_1} & s_{g_1} & s_{b_1} & s_{r_1}^2 & s_{g_1}^2 & s_{b_1}^2 & s_{r_1}^3 & s_{g_1}^3 & s_{b_1}^3 \\ s_{r_2} & s_{g_2} & s_{b_2} & s_{r_2}^2 & s_{g_2}^2 & s_{b_2}^2 & s_{r_2}^3 & s_{g_2}^3 & s_{b_2}^3 \\ \vdots & \vdots & \vdots & \vdots & \vdots & \vdots & \vdots & \vdots & \vdots \\ s_{r_{24}} & s_{g_{24}} & s_{b_{24}} & s_{r_{24}}^2 & s_{g_{24}}^2 & s_{b_{24}}^2 & s_{r_{24}}^3 & s_{g_{24}}^3 & s_{b_{24}}^3 \end{bmatrix} \quad (2)$$

3. Calculate $M$, the Moore–Penrose inverse matrix of $S$ (* denotes transpose).

$$M = (S^*S)^{-1}S^* \quad (3)$$

4. Estimate standardization vectors of each R, G, and B channel by multiplying $M$ with each column of $T$.

$$\begin{bmatrix} R_h \\ G_h \\ B_h \end{bmatrix} = \begin{bmatrix} MT_R \\ MT_G \\ MT_B \end{bmatrix} = \begin{bmatrix} \hat{r}_r & \hat{r}_g & \hat{r}_b & \hat{r}_{r^2} & \hat{r}_{g^2} & \hat{r}_{b^2} & \hat{r}_{r^3} & \hat{r}_{g^3} & \hat{r}_{b^3} \\ \hat{g}_r & \hat{g}_g & \hat{g}_b & \hat{g}_{r^2} & \hat{g}_{g^2} & \hat{g}_{b^2} & \hat{g}_{r^3} & \hat{g}_{g^3} & \hat{g}_{b^3} \\ \hat{b}_r & \hat{b}_g & \hat{b}_b & \hat{b}_{r^2} & \hat{b}_{g^2} & \hat{b}_{b^2} & \hat{b}_{r^3} & \hat{b}_{g^3} & \hat{b}_{b^3} \end{bmatrix} \quad (4)$$

5. To standardize the R, G, and B components of each pixel, $i$, in the source image, apply each standardization vector to $S_i$, the linear, quadratic, and cubic RGB components of $i$.

$$S_{i,standardized} = \begin{pmatrix} S_i R_h & S_i G_h & S_i B_h \end{pmatrix}$$

$$\text{where} \quad S_i = \begin{pmatrix} r_i & g_i & b_i & r_i^2 & g_i^2 & b_i^2 & r_i^3 & g_i^3 & b_i^3 \end{pmatrix}. \quad (5)$$

The resulting image will have the same color profile, contrast, and brightness of the target image. To quantify the deviance, denoted as $D$, between the source and target images, extend $T$ to include quadratic and cubic terms for each R, G, and B exactly as (2) and estimate standardizations for the six new columns in $T$.

6. Construct matrix $H$ using all nine standardizations vectors.

$$H_{9x9} = \begin{bmatrix} R_h \\ G_h \\ B_h \\ R_h^2 \\ G_h^2 \\ B_h^2 \\ R_h^3 \\ G_h^3 \\ B_h^3 \end{bmatrix} = \begin{bmatrix} MT_R \\ MT_G \\ MT_B \\ MT_{R^2} \\ MT_{G^2} \\ MT_{B^2} \\ MT_{R^3} \\ MT_{G^3} \\ MT_{B^3} \end{bmatrix}. \tag{6}$$

7. Compute deviance from reference.

$$D = 1 - \det(H). \tag{7}$$

If the source image has an identical color profile as the target image, $H$ is the identity matrix and the matrix determinant is 1. So that no change in profile is reported as a 0, one minus the determinant is the returned value.

## RESULTS

### Identifying previously unknown source of variance

The Bellwether Phenotyping Facility at the Donald Danforth Plant Science Center is a controlled-environment growth facility that utilizes a Scanalyzer 3D-HT plant-to-sensor system (LemnaTec GmbH) to move plants through stationary imaging cabinets (*Fahlgren, Gehan & Baxter, 2015*). The design of the imaging cabinets is intended to produce consistent images with uniform backgrounds. To test the consistency of images collected by the Bellwether system, we imaged sorghum plants for 11 days and included a ColorChecker Passport panel in the field of view so image color profiles could be compared. We designated an image taken at noon on the first day of the experiment to be the reference, and we calculated the deviance ($D$) for every other image. We observed that there was large variance in $D$ in the image set (median $= -0.370$, mean $= -8.756$, variance $= 530.991$). By manually inspecting images that showed a large and small $D$, we observed that a large, negative $D$ corresponded to a decrease image brightness. Furthermore, we observed that $D$ had a periodic distribution with the time of day the image was taken. Collapsing all the images taken over the 11-day span to a 24-hour window shows that there is a clear time of day effect (Fig. 1A). Images taken during experimental nighttime hours had decreased brightness compared to images taken during daytime hours. (Wilcoxon rank-sum test, $p$-value $< 0.001$). We tested two hypotheses that could account for the observed diurnal pattern: (1) light leakage into the imaging cabinet through its vents during the day, or (2) the temperature of the room was influencing the brightness of the fluorescent bulbs that illuminate the cabinet. To test the former hypothesis, we took two pictures containing a ColorChecker Passport panel: one with the vents open and another with the vents covered. Setting one of images as the reference and standardizing the other to it resulted

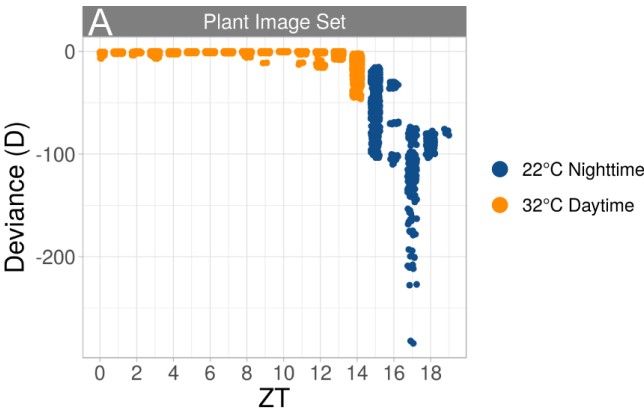

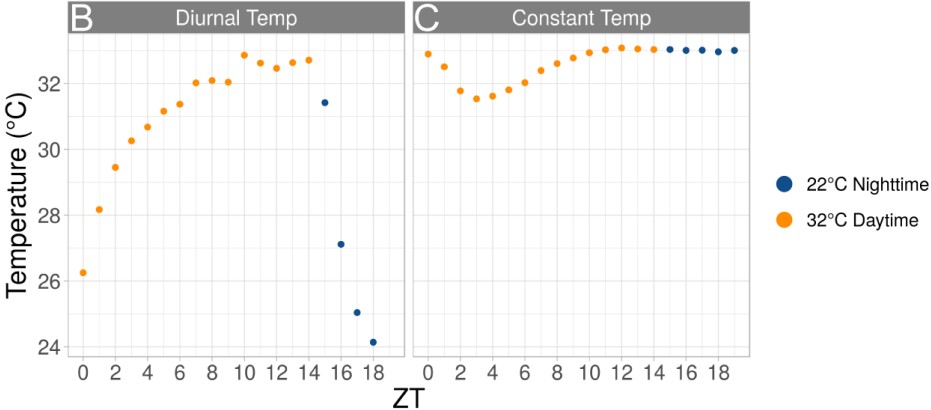

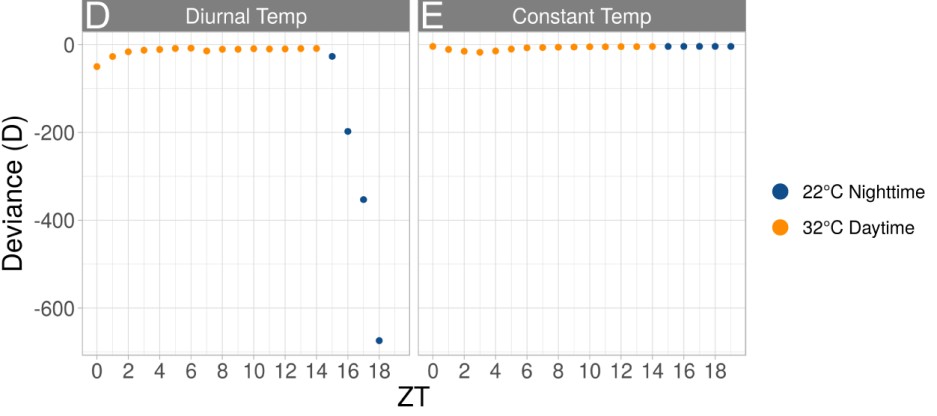

**Figure 1  Effect of temperature on light intensity and image variability.** (A) Graph of calculated deviance ($y$-axis) within the plant image set and the hour the image was taken in ZT time ($x$-axis). Each dot represents a single image in the plant image set. The deviance from reference was calculated in Eq. (7). (B and C) Graphs of temperature readings ($y$-axis) taken at hours at which imaging was done in ZT time ($x$-axis) in either diurnal (B) or constant (C) conditions. (D and E) Graphs of calculated deviance ($y$-axis) within the temperature image set and the hour the image was taken in ZT time ($x$-axis) under either diurnal (D) or constant (E) conditions. Each dot represents a mean of two images in the "temperature image set". The deviance from reference was calculated in Eq. (7). (A–C). Daytime ZT 0–14 = hr 0,700–2,100 and nighttime ZT 14–24 = hr 2,100–0,700 (A–E). Temperature settings at particular times of day are indicated, either 22 °C (blue) or 32 °C (gold).

in $D = -0.454$, which is insufficient to explain the large absolute $D$ values observed in the plant image set. To test the latter hypothesis, a two-day test was done with a 14 hr/10 hr (day/night) photoperiod. For the first 24 hrs the temperature cycled with the photoperiod (32 °C/22 °C, day/night). For the second 24 hrs temperature was held at a constant 32 °C. Five thermometers were placed throughout the imaging cabinet and recorded temperature every minute. Images containing a ColorChecker Passport panel were captured every hour (ZT0-ZT18; to mimic the period of time images were collected in the full experiment) and used to calculate $D$ for each image. Temperatures within the chamber tracked closely with those in the room (Figs. 1B and 1C). We observed that under diurnal conditions, $D$ varies with temperature, and in particular, large absolute values of $D$ were observed for temperatures below 28 °C (Fig. 1D). Additionally, the variation of $D$ was consistent with the variation observed for the plant image set both in frequency and magnitude (Fig. 1). Furthermore, when the ambient temperature remained constant, $D$ did not change (Fig. 1E). The imaging cabinet is illuminated with fluorescent bulbs and it is known that light output from this type of bulb is affected by ambient temperature (*Bleeker & Veenstra, 1990*). Using the measure of deviance defined in Eq. (7), we were able to detect and identify a source of variance within our plant imaging facility.

## Standardization of images improves shape and color measurements

Given that there was variable brightness in the plant image set, the commonly used fixed-threshold segmentation routines produced noisy results. Dark, low-quality images that were segmented using a fixed-threshold routine optimized for bright, high-quality images resulted in either background pixels being classified as object, object pixels being classified as background or both (Figs. 2A–2D). For example, segmentation of an uncorrected sample image take during the experimental night fails to remove a large amount of background near darker parts of the image and salt and pepper noise around the plant. In contrast, segmentation of the transformed image results in clean segmentation of the plant without any additional filtering needed.

To further explore how our standardization method affects phenotype analysis, a set of morphological shapes of each plant was measured before and after standardization. We calculated the difference between the measurements obtained from the original images and the standardized images and grouped the results into experimental day and night (Fig. 2E). The values of the shapes are on vastly different scales so normalization of those values by subtracting the grand mean and dividing by the variance was done for each image and for each shape. The effects in this scale are equivalent to the effects in the native scale for each shape. Many commonly reported measurements (area, convex hull area, width, perimeter, circularity, solidity, center of mass $x$-coordinate, height, ellipse minor axis, fractal dimension, roundness, eccentricity, center of mass $y$-coordinate, ellipse angle, convex hull vertices) are significantly different using an unequal variance $t$-test ($p$-value < 0.05) between images taken during daytime and nighttime temperatures. Other common measurements such as ellipse major axis, ellipse center $x$-coordinate, aspect ratio, and ellipse center $y$-coordinate were not significantly different (Fig. 2E). Plant area (cm$^2$) was previously shown to correlate to fresh weight biomass on the Bellwether platform (*Fahlgren,*

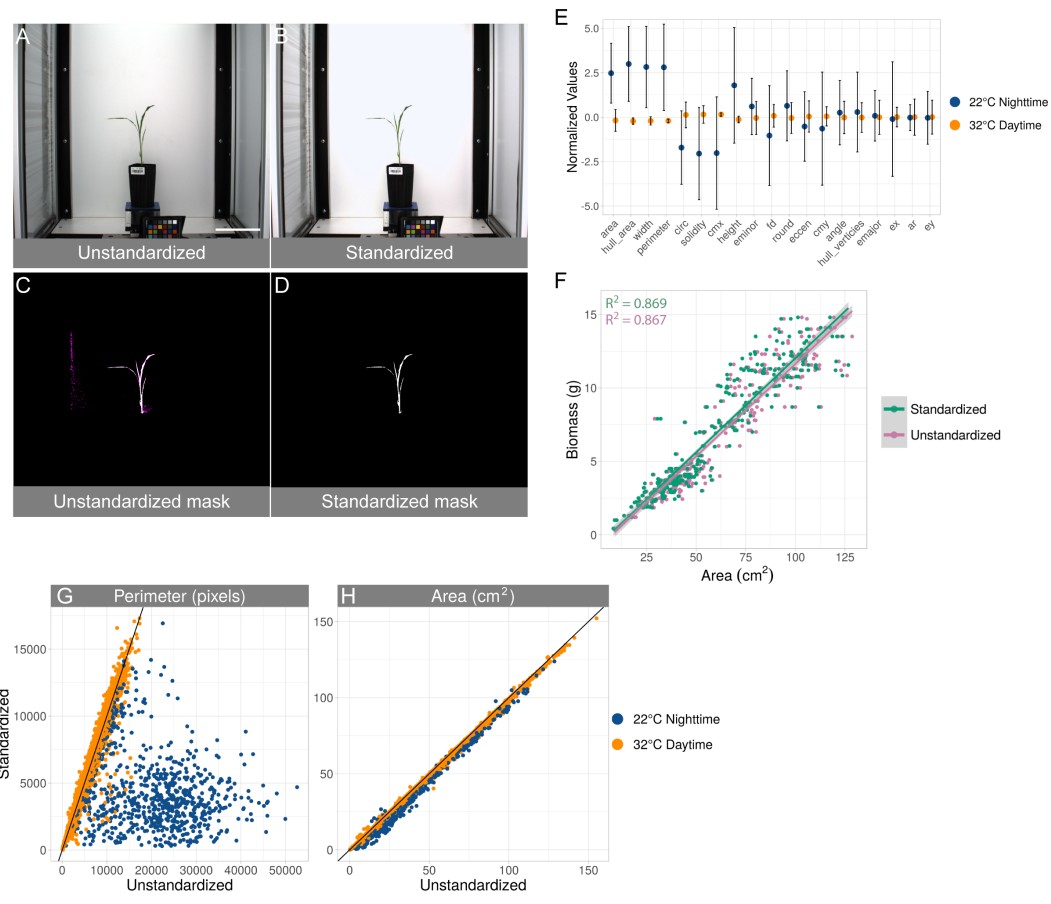

**Figure 2** **Standardization effects on shape measurements.** (A–D) Example of the effect of standardiza-
tion. (A) original unedited image. (B) same image after applying standardization. Plant-isolation masks
generated without (C) and with (D) standardization. The magenta coloring in the unstandardized mask
indicates pixels that are not contained in the standardized mask. Scale bar = 15 cm. (E) Effect size ($y$-axis)
of each of the shapes ($x$-axis) before and after standardization grouped by experimental day and night.
Values of each shape per image are normalized by subtracting the mean and dividing by the variance. All
shapes except ellipse major axis, ellipse center $x$-coordinate, aspect ratio, and ellipse center y-coordinate
are significantly different between daytime and nighttime temperatures (Unequal variance $t$-test, $p$-value
< 0.05). Shapes are sorted by increasing $p$-values and are: area, convex hull area, width, perimeter, circu-
larity, solidity, center of mass $x$-coordinate, height, ellipse minor axis, fractal dimension, roundness, ec-
centricity, center of mass $y$-coordinate, ellipse angle, convex hull vertices, ellipse minor axis, ellipse center
$x$-coordinate, aspect ratio, and ellipse center y-coordinate. (F) Association of biomass ($y$-axis) to area ($x$-
axis) before and after standardization. Every point is an image taken on the last day within the plant im-
age set for which weight was recorded. Linear fit (green line: $y = -0.69 + 0.126\times$; pink line: $y = -0.85 +
0.124\times$) and $R^2$ is displayed for each condition. (G and H) Effect of standardization on perimeter (G) and
area (H). Every point is an image in the entire plant image set. Displayed are the measured perimeter and
area from before standardization ($x$-axis) and after standardization ($y$-axis). Temperature settings at par-
ticular times of day are indicated, either 22 °C (blue) or 32 °C (gold). Black line indicates $y = x$.

*Gehan & Baxter, 2015*), and we observed a similar relationship in the current experiment
with a small subset of plants ($n = 162$) that were manually weighed on the last day of the
experiment (Fig. 2F). Color profile standardization had no effect on fresh-weight biomass
prediction as the slopes and intercepts of the linear models using the unstandardized and

standardized datasets were not significantly different ($p$-value > 0.05, two-way ANOVA). Plant area responds to salt and pepper noise in a one-to-one relationship, so a relatively small amount of noise compared to total plant area will not produce large effects. In contrast, shapes like perimeter are more sensitive and produce larger differences because inclusion of one background pixel can add one or more pixels to the resulting measurement (Figs. 2G and 2H).

Next, we sought to understand how the color profile of the object was affected by our standardization method. Hue is a measure of the color in degrees around a circle of an object in an image, and we previously used hue to measure plant responses to abiotic stress (*Moroney et al., 2002*; *Veley et al., 2017*). We compared the hue profiles of plants from original and standardized images and observed an increase in signal above 120° (green) in the nighttime temperature unstandardized images relative to the daytime temperature unstandardized images (Figs. 3A and 3C). Additionally, we observed a large amount of noise below 90° (red-yellow) in the same comparison. Visual inspection of a subset of images suggested that the former is due to misinterpretation of the true object pixels' shade of green and the latter is due to background being included in the hue histogram. The standardized image sets result in consistent color profiles between the nighttime and daytime images (Figs. 3B and 3D). To statistically test for differences between hue histograms, a two-sample Kolmogorov–Smirnov test was applied to each pairwise comparison. Hypothesis testing was done using the following:

$$K > c(\alpha)\sqrt{\frac{n+m}{nm}} \tag{8}$$

$$c(\alpha) = \sqrt{-\frac{1}{2}\ln(\frac{\alpha}{2})} \tag{9}$$

$$K = \sup\left|F_x(\text{hue}) - F_y(\text{hue})\right| \tag{10}$$

$F$ is a cumulative distribution of samples $x$ and $y$, each with sample sizes $n$ and $m$, respectively. In Eq. (8), if the left-hand side is greater than the right-hand side at $\alpha = 0.05$, then we reject the null hypothesis that the two histograms are the same distribution. Using this test, we determined that the standardization procedure significantly alters the color profile of the images taken under nighttime temperatures but not the images taken under daytime temperatures, and the standardized image set are no longer significantly different between day and night temperatures (Fig. 3E). This suggests that the standardization method accurately maps the color space from the dark, low-quality images to the same color space as the bright, high-quality images. Changes in hue spectra have been shown to be associated to varying levels of nitrogen in sorghum and rice (*Veley et al., 2017*; *Wang et al., 2014*). We hypothesized that variation in color profiles due to differences in light quality would affect our ability to accurately measure nitrogen treatment effects in the present dataset. We used a constrained analysis of principal coordinates conditioned on time, in days, to determine whether treatment groups could be separated by hue values (Figs. 3F–3I). Consistent with previous results, clear separation of the nitrogen treatment

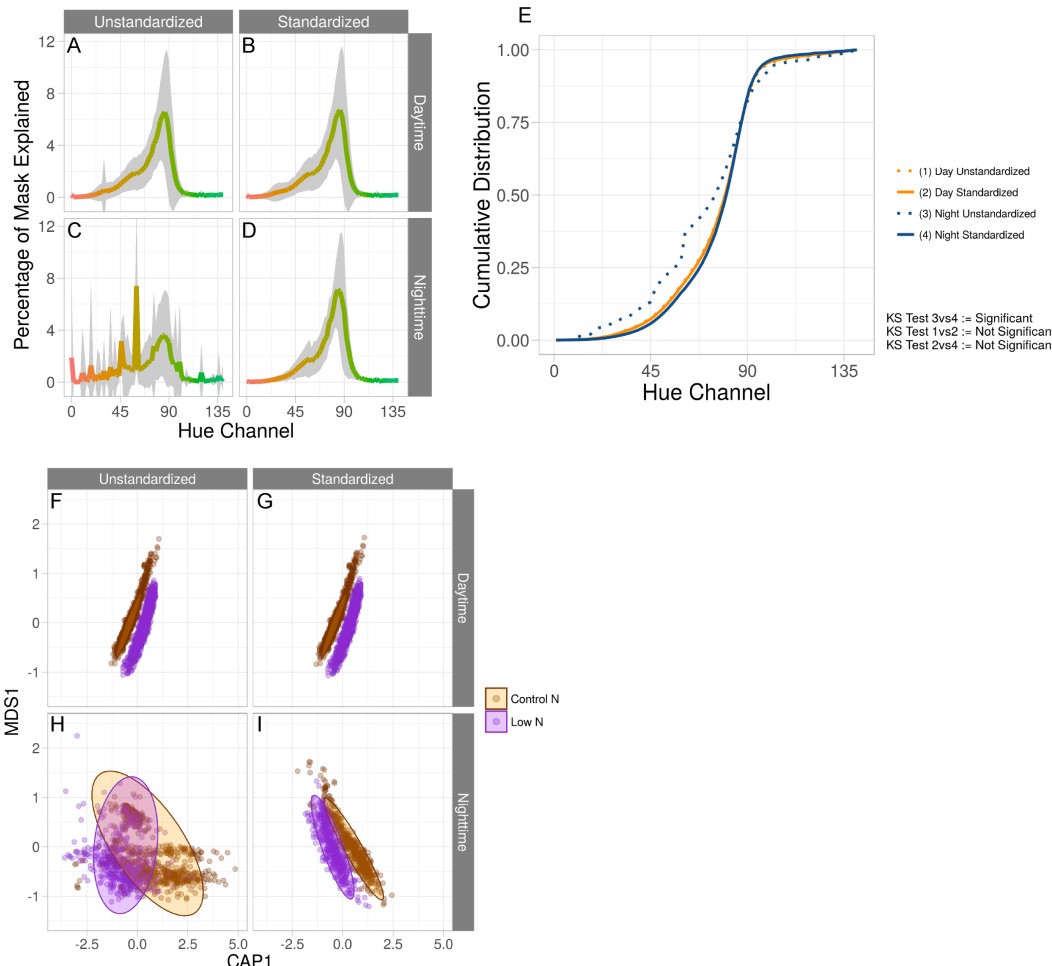

**Figure 3** **Standardization effects on color measurements.** (A–D) Average hue histograms of plants 19 days after planting. Shown are the average histograms of unstandardized daytime and nighttime images as well as standardized daytime and nighttime images. For each facet, the *x*-axis is the hue channel from 0 to 140 degrees and the *y*-axis is the percentage of the mask at a particular degree. The color of each degree is shown as the color of the line. Gray areas are the 95% confidence interval. (E) Using the same histograms as in (A–D), cumulative distributions of each are shown on the *y*-axis, and the hue range from 0 to 140 are on the *x*-axis. As the legend indicates, each line is given a number and the Kolmogorov–Smirnov test as described in Eqs. (8), (9) and (10) are displayed using the numbers that are assigned to each condition. (F–I) Constrained analysis of principal coordinates conditioning on time, in days, for both unstandardized and standardized images colored by nitrogen treatment for both daytime and nighttime images. Constrained axis 1 (CAP1, *x*-axis) is conditioned on time while multidimensional scaled axis 1 (MDS1, *y*-axis) is not conditioned on any variables.

groups was observed for both unstandardized and standardized images taken during the daytime (Figs. 3F and 3G; *Veley et al., 2017*; *Wang et al., 2014*). In contrast, unstandardized nighttime images appeared to have relatively similar hue spectra across nitrogen treatments, whereas standardized nighttime images show clear separation (Figs. 3H and 3I).

## DISCUSSION

Variation within an image-based dataset can inhibit the accurate measurement of shape-based attributes. Additionally, color profiling is important for understanding plant disease, chlorophyll content, and transpiration rate (*Mutka & Bart, 2015*) and these phenotypes would likely be affected by image set variation similar to what we observe here. Although the image variation described here was specifically found in the Bellwether Phenotyping Facility at the Donald Danforth Plant Science Center, it serves as an example of the potential for variability in even tightly controlled imaging systems. In many cases, imaging conditions may be far less controlled, contributing to image variability and measurement error. In our case, the source of variation could potentially be removed by using a different light source such as LEDs, although these have been shown to have temperature-dependent light output as well (*Narendran et al., 2007*). Regardless, it is often impractical to remove all sources of variation, so having a methodical way to standardize an image set allows anyone using image analysis to reduce the variance of the numerical output. Reducing variance will increase statistical power, allowing for observation and inference of smaller effects that would otherwise not be quantifiable. While it might not be possible to include color reference cards in every image for all applications, our results suggest that it is an important consideration for plant phenotyping systems, and a key aspect of our method is that it can be used as a tool to assess whether significant variability exists by doing a relatively straightforward pilot experiment.

In addition, there are many references to choose from and the choice of reference matters (*Ilie & Welch, 2005*). We chose ColorChecker Passport Photo (X-Rite, Inc., Grand Rapids, MI, USA), a commercially available 24-chip color card as a reference, because it contained a diverse array of colors while being relatively compact. During the analysis process, we tried removing random reference colors from the reference and determined that we could remove as many as 8 chips before the transformation matrix applied to the images became incalculable. However, the minimum and optimum requirements of a color standard are likely to be dataset-, camera- and light source-dependent. Determining the exact requirements of the color standard will be the focus of future work.

It is important to note that there are other methods that exist to solve the problem of segmenting objects from an image set of varying quality. Some methods include Bayes classifiers, neural networks, and autothresholders (*Yogamangalam & Karthikeyan, 2013*). All object segmentation methods aim to produce a binary image wherein all object pixels are denoted as 1 and all background are denoted 0. Autothresholders attempt to alleviate misclassification by varying the value of the threshold depending on other characteristics of the image. Neural networks and Bayes classifiers work in a similar fashion by learning what is object and what is background and making decisions based on those models. While these methods have been proven to show better classification of object to background vs fixed-threshold methods, these methods do not change the values of the pixels, so standardized color analysis would still be problematic under variable conditions. Another downside to methods such as these is that they require training data from across the spectrum of observed variation so that robust classification can be done. Instead, we

propose that image standardization is a key first step in image analysis that will simplify the computational complexity of downstream steps, regardless of choice in segmentation method, and will improve the reliability of results and inferences.

## CONCLUSIONS

We show that using the image standardization method described here improves fixed-threshold object segmentation and creates an environment where detailed color measurements are possible. Improving segmentation means shape measurements of the object are more accurate to physical, ground truth measurements. This standardization method also changes the values of the pixels of in the image, and we show it accurately maps the color space from a given image to a high-quality reference image. Finally, this method supplies a measure of deviation from an image to a reference and can be used to identify sources of variance in an image set. Future research will determine the limits of this method of standardization but it is not likely to be limited to high-throughput plant phenotyping and will be applicable to many image-based experiments.

## ACKNOWLEDGEMENTS

We acknowledge and thank Mindy Darnell and Leonardo Chavez from The Bellwether Foundation Phenotyping core facility at the Danforth Center for maintaining the imaging facility. We also acknowledge and thank Molly Kuhs for her assistance in setting up and maintaining the reagents needed for the experiment and Dr. Stephen Kresovich for his work in supplying the seed.

### Funding

This work was supported by the Donald Danforth Plant Science Center, the US National Science Foundation (IIA-1355406, IIA-1430427, and DBI-1659812), the US Department of Energy (DE-AR0000594, DE-SC0014395, and DE-SC0018072), and the US Department of Agriculture (2016-67009-25639). The funders had no role in study design, data collection and analysis, decision to publish, or preparation of the manuscript.

### Grant Disclosures

The following grant information was disclosed by the authors:
Donald Danforth Plant Science Center.
US National Science Foundation: IIA-1355406, IIA-1430427, DBI-1659812.
US Department of Energy: DE-AR0000594, DE-SC0014395, DE-SC0018072.
US Department of Agriculture: 2016-67009-25639.

### Competing Interests

Jeffrey C. Berry, Noah Fahlgren, Alexandria A. Pokorny, Rebecca S. Bart, and Kira M. Veley contributed to the research described while working at the Donald Danforth Plant Science Center, a 501(c)(3) nonprofit research institute.

## Author Contributions

- Jeffrey C. Berry and Kira M. Veley conceived and designed the experiments, performed the experiments, analyzed the data, contributed reagents/materials/analysis tools, prepared figures and/or tables, authored or reviewed drafts of the paper, approved the final draft.
- Noah Fahlgren and Rebecca S. Bart conceived and designed the experiments, contributed reagents/materials/analysis tools, authored or reviewed drafts of the paper, approved the final draft.
- Alexandria A. Pokorny contributed reagents/materials/analysis tools, authored or reviewed drafts of the paper, approved the final draft.

## Data Availability

This method is available for anyone to use and comes in two forms.

The first is the C++ source code that is supplied with this paper and must be compiled against OpenCV (only tested against version 3.1). This program is licensed with GNU General Public License v2. The code is deposited at GitHub https://github.com/jberry47/phenotypercv.

The second is a submodule that comes with PlantCV (versions >3.0.dev1). PlantCV is available on GitHub at https://github.com/danforthcenter/plantcv. PlantCV v3.0.dev1 is archived on Zenodo at https://doi.org/10.5281/zenodo.1296371. PlantCV is licensed with MIT Open Source Initiative. The PlantCV Public image datasets database is https://plantcv.danforthcenter.org/pages/data.html.

Original images, corrected images, R scripts and all numerical data are available to download at https://bioinformatics.danforthcenter.org/phenotyping/. Images are licensed with Creative Commons v1.0. R scripts and corresponding data files are provided in the Supplemental File and can be used to reproduce all graphs and statistics.

## Supplemental Information

Supplemental information for this article can be found online at http://dx.doi.org/10.7717/peerj.5727#supplemental-information.

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
