# Peer review of "An automated, high-throughput method for standardizing image color profiles to improve image-based plant phenotyping"

_PeerJ, doi:10.7717/peerj.5727_

## Round 0.1 · original submission · Minor Revisions

Two of three reviewers suggest minor revision. I believe it will take a short time to update the manuscript. Please consider comments and suggestion from the Reviewer 1. It will help in the paper presentation. Waiting your revised manuscript.

Reviewer 1 ·

Basic reporting

no comment

Experimental design

no comment

Validity of the findings

no comment

Additional comments

In the work of Jeffrey C Berry and co-authors the high-throughput method for standardizing image color profiles to improve image-based plant phenotyping is presented.

The authors used the Scanalyzer 3D-HT plant-to-sensor system for obtaining digital images of plants. A ColorChecker Passport Photo with a palette of 24 colors was placed in the area of the frame, which was used for color correction by the method described in Gong et al., 2016 and described in MATERIALS AND METHODS, Standardization Method. For each image, the authors estimated the variance of the observed colors from the reference values and obtained several interesting results:
Dispersion for images obtained at night was higher than for images obtained in the daytime. The authors showed that the differences are due to the use of a fluorescent lamp, in which the luminous flux varies with a lower temperature at night.
The authors calculated a number of quantitative characteristics on the source images and on the images after correction. It was shown that in the calculation of some features (perimeter, height), there is a strong variability in the case of using unstandardized images. This is due to poor-quality detection of the plant contour on the unstandardized image.
The authors constructed a hue histogram for standardized and unstandardized images obtained at day and night. It was shown that the histograms of unstandardized images, made during day and night, have significant statistical differences.
In conclusion, because the color profile of the leaves of the plant varies with the concentration of nitrogen, the authors performed clustering of standardized and unstandardized images obtained at night. It was shown that the separation, depending on the nitrogen concentration, is observed for the group of standardized images.

There are a number of remarks to the paper:
1. The article does not explain why the correction method (Gong et al., 2016) was chosen and no comparison is made with other existing methods. There are works in which it is shown that the transition from the RGB color space to the XYZ space can improve the quality of the correction.
Mackiewicz M., Andersen C. F., Finlayson G. Method for hue plane preserving color correction //JOSA A. – 2016. – Т. 33. – №. 11. – С. 2166-2177.
Quintana J., Garcia R., Neumann L. A novel method for color correction in epiluminescence microscopy //Computerized Medical Imaging and Graphics. – 2011. – Т. 35. – №. 7-8. – С. 646-652.
2. Figures 1B and 1D show that as the dispersion of observed colors from the control values increases, the scatter of the measured morphological characteristics varies in different ways. It would be nice to formulate a clear statistical criterion about which morphological characteristics can be calculated without applying the color correction algorithm, and which ones are not.
3. In Figure 1C, the differences between the standardized and unstandardized images are not visually discernible. The correlation coefficients are practically the same. It is necessary to supplement this figure with linear regression equations (the slope and intercept parameters). It would be interesting to compare these parameters between day and night images. Are they differ significantly?
4. For Figure 3C, it is interesting to build variants of clustering for images obtained in the daytime (standardized and not corrected) to make sure that the results will be the same as we see on 3A.

There are several wishes for the work:
1. It is interesting to measure how to change the accuracy of calculating various morphological features when using ColorChecker with 16 colors instead of 24.
2. To detect the plant contour, a threshold segmentation algorithm is used. It is interesting to measure how to change the accuracy of calculating various morphological features using the segmentation method with the adaptive threshold or other more "smart" methods that were mentioned in the DISCUSSION section.
3. The validation of the measured quantitative characteristics of plants does not occur in any way, except for weighing ~ 120 plants. At the same time, if the hardware system used can produce a 3D model of the plant, then it could be used for additional validation of the color correction model proposed by the authors.

·

Basic reporting

The paper is written in clear and concise English with only minor errors in grammar and punctuation (i.e. a few missing periods and the parenthetical annotation of references is inconsistent). All of the supplementary data provided could be opened and accessed easily.

The following two items should be corrected

1. Importantly, the size of the font is too small for the keys in Fig. 2B, 2C, 2D, 3B, and 3B should be enlarged for clarity of the figure panels. I would recommend correcting this.
2. Based on the figures and materials and methods, the day length should be 14 hours with 10 hours of night (noted ZT 2100-0700), but is noted as a 9 hr night in the materials and methods. This minor correction should be made.

Experimental design

The experimental design is well planned and carried out using robust and well documented statistical analyses. The experiments described and carried out in the manuscript point out an important constraint in phenotypic analyses and point both to a causation of the color variation and a solution for correction in the standardization method outlined in the manuscript.

Validity of the findings

Moreover, the authors provide not only validation of their image standardization method, but also provide an example of improved resolution of N-treatment effects on plant phenotype by using standardization. Thus, they provide strong evidence that color standardization is necessary in image analyses for htp phenotyping data.

Additional comments

Color correction is not only a problem in controlled environment phenotyping platforms but also in field-based systems with fluctuating light. For example, color correction panels are routinely used for field phenomics for crop plants. A recent report : Chopin et al., 2018 Plant Methods “Land-based crop phenotyping by image analysis: consistent canopy characterization from inconsistent field illumination” also describes these issues. The color standardization documented in the current manuscript would be of benefit and perhaps quite useful for field-based phenomic studies.

·

Basic reporting

In this paper Berry et al. report a color correction method that improves image segmentation for a number of image-based features often extracted for phenotypic analysis in plants. This tactic for improving segmentation is straight-forward and seems to this reviewer to be a method that nearly all plant phenotypers will want to incorporate into their pipelines to improve object segmentation.

Comment: could figure 2, panel A be zoomed in so that the pic is primarily of the plant itself? This would help with any potential to discern a difference between the bottom two masked panels.

Data: I see that the input image datasets are available. Are the corrected datasets available? It would be nice to be able to try to reproduce the latter based upon direct implementation of the methods and applying those to the former.

Experimental design

Well done. I am impressed by the both the biological experimental design as well as the computational experimental design.

Validity of the findings

no comment

Additional comments

Nice manuscript. Very timely and useful.

MINOR ITEM:
Spelling error. Palate is a feature of taste or an anatomical feature of the mouth. Palette is the color selection.

---

## Round 0.2 · accepted · Accept

The paper has no more remarks from the reviewers.

# Reviewer 1 ·

Basic reporting

no comment

Experimental design

no comment

Validity of the findings

no comment

Additional comments

no comment

·

Basic reporting

no comment

Experimental design

no comment

Validity of the findings

no comment

Additional comments

Revisions have rendered this manuscript acceptable for publication.